# Primary Healthcare Physicians’ Insufficient Knowledge Is Associated with Antibiotic Overprescribing for Acute Upper Respiratory Tract Infections in China: A Cross-Sectional Study

**DOI:** 10.3390/antibiotics13100923

**Published:** 2024-09-26

**Authors:** Muhtar Kadirhaz, Yushan Zhang, Naveel Atif, Wenchen Liu, Wenjing Ji, Nan Zhao, Jin Peng, Sen Xu, Miaomiao Xu, Chengzhou Tang, Yu Fang, Jie Chang

**Affiliations:** 1Department of Pharmacy Administration, School of Pharmacy, Xi’an Jiaotong University, Xi’an 710061, China; muhtar@stu.xjtu.edu.cn (M.K.); zys20170521@stu.xjtu.edu.cn (Y.Z.); atifnaveel@stu.xjtu.edu.cn (N.A.); liuwenchen@stu.xjtu.edu.cn (W.L.); wenjing.ji@xjtu.edu.cn (W.J.); a528434146@stu.xjtu.edu.cn (N.Z.); penjin21@stu.xjtu.edu.cn (J.P.); marsxs@stu.xjtu.edu.cn (S.X.); mm_xu@stu.xjtu.edu.cn (M.X.); 3123315106tcz@stu.xjtu.edu.cn (C.T.); 2Center for Drug Safety and Policy Research, Xi’an Jiaotong University, Xi’an 710061, China

**Keywords:** primary healthcare, antibiotic use, upper respiratory tract infections

## Abstract

Objectives: Overuse of antibiotics in healthcare remains prevalent and requires urgent attention in China, particularly in primary healthcare (PHC) facilities. This study aimed to describe the patterns of antibiotic prescriptions for acute upper respiratory tract infections (URTIs) in PHC facilities in China and to investigate how PHC physicians’ knowledge influences their antibiotic prescribing behavior. Methods: A cross-sectional survey linking physician questionnaire responses and their prescription data was conducted in Shaanxi Province, China. The proportions of URTI visits that received at least one antibiotic, combined antibiotics, and broad-spectrum antibiotics were the main outcomes reflecting antibiotic prescribing behavior. Multivariate mixed-effects logistic regressions were applied to analyze the relationship between PHC physicians’ knowledge about antibiotics and their antibiotic prescribing behavior. Results: A total of 108 physicians filled out the questionnaires between February 2021 and July 2021, and a sample of 11,217 URTI visits attended by these physicians from 1 January 2020 to 31 December 2020 were included in the analysis. The overall mean score of our respondents on the knowledge questions was 5.2 (total score of 10). Over sixty percent (61.2%; IQR 50.2–72.1) of the URTI visits received antibiotics. The percentages of URTI visits prescribed with combined and broad-spectrum antibiotics were 7.8% (IQR 2.3–10.2) and 48.3% (IQR 36.7–58.7), respectively. Third-generation cephalosporins were the most frequently used antibiotics. Physicians with lower antibiotic knowledge scores were more inclined to prescribe antibiotics (*p* < 0.001), combined antibiotics (*p* = 0.001), and broad-spectrum antibiotics (*p* < 0.001). Conclusions: Physicians’ insufficient knowledge was significantly associated with antibiotic overprescribing. Professional training targeting physicians’ knowledge of antibiotics is urgently needed to improve the rational use of antibiotics in grassroots healthcare facilities in China.

## 1. Introduction

Antimicrobial resistance (AMR) is recognized as an urgent global public health problem [1]. It is estimated that AMR was directly responsible for 1.27 million deaths worldwide and associated with 4.95 million deaths in 2019 [2]. The misuse and overuse of antibiotics significantly contribute to the development of AMR [3]. Optimizing antibiotic use is considered a core measure to address this global threat [1].

China is one of the world’s largest consumers of human antibiotics [4]. Previous studies have shown that antibiotics are frequently prescribed in Chinese primary healthcare (PHC) facilities, with a large proportion of these prescriptions being inappropriate [5]. The most common reasons for visits to PHC facilities are acute upper respiratory tract infections (URTIs), which usually result in antibiotic prescriptions [6]. However, most acute URTIs are self-limiting, and antibiotics do not shorten the duration of the illness or prevent complications [7]. A nationwide study revealed that 55.1% of acute URTI patients were prescribed antibiotics at PHC facilities in China [6]. Therefore, studies analyzing the patterns and driving factors of antibiotic prescriptions for acute URTIs in PHC settings are critical for informing efforts to curb the overuse of antibiotics.

Physicians play a critical role in optimizing the use of antibiotics. However, their prescribing behavior is complex and can be influenced by various factors. Previous studies have revealed associations between physicians’ antibiotic prescribing behavior and several external factors, such as time constraints, patient demands, financial incentives, and organizational settings [8,9]. Nevertheless, medical knowledge regarding antibiotics, as an essential internal factor, has been recognized as a prerequisite for appropriate antibiotic prescribing [10]. The higher the physicians’ knowledge of antibiotics, the greater the likelihood that the antibiotic prescription would be rational [11].

Despite the importance of knowledge, previous studies that examined the role of PHC physicians’ knowledge in their antibiotic prescribing practices have shown inconsistent findings. While researchers in Spain found that the prescribing of antibiotics was closely associated with the specific knowledge of PHC physicians, studies conducted in Scotland and China revealed that PHC physicians’ higher knowledge failed to translate into better antibiotic prescribing practices [12,13,14]. Given the substantial proportion of PHC physicians in China with limited educational backgrounds and insufficient professional training [15], it is imperative to gather more empirical evidence to thoroughly examine and understand the impact of knowledge on antibiotic prescribing among PHC physicians in China.

The aims of the current study were twofold: firstly, to describe the patterns of antibiotic prescriptions for acute URTIs in PHC facilities in Shaanxi Province, China; and secondly, to analyze how current antibiotic prescribing for URTI patients is influenced by PHC physicians’ antibiotic knowledge. We believe that the findings from this study will be essential for designing effective interventions to optimize antibiotic use in PHC settings.

## 2. Results

Of 119 eligible physicians practicing at the 30 PHC facilities during the study period, a total of 108 completed the questionnaire, yielding a response rate of 91%. A total of 11,217 URTI visits attended by these 108 physicians were included in the analysis.

### 2.1. Physician Characteristics and Antibiotic Prescribing Patterns

The demographic details of the 108 participating physicians are summarized in Table 1. The number of female physicians (55.6%) was higher than that of male physicians (44.4%). Most physicians (46.3%) were between 36 and 50 years old. Slightly over half of the participants were from THCs. Sixty-one (56.5%) physicians had a bachelor’s degree or higher. Around thirty percent (29.6%) of the respondents had a working experience of over 25 years. Nearly half (49.1%) of our respondents were attending physicians. Regarding the knowledge of antibiotics, 54.6% of the physicians gave correct answers to less than six questions.

Of the 11,217 URTI visits to the study physicians, over sixty percent (61.2%; IQR 50.2–72.1) were prescribed at least one antibiotic. The percentage of visits prescribed with combined and broad-spectrum antibiotics was 7.8% (IQR 2.3–10.2) and 48.3% (IQR 36.7–58.7), respectively. Prescribing patterns of different physician subgroups are listed in Table 1, and details of URTI visits included in the analysis are presented in Appendix A.

In total, 7797 individual antibiotics were prescribed for sampled URTI visits, with 5859 (75.1%) being broad-spectrum antibiotics (Table 2). The most commonly prescribed antibiotics were third-generation cephalosporins (34.7%), followed by penicillins (29.5%), second-generation cephalosporins (11.0%), macrolides (8.5%), and fluoroquinolones (7.6%). As for specific agents, cefixime (27.1%) and cefuroxime (8.5%) were the most commonly prescribed cephalosporins, while amoxicillin (16.5%) and amoxicillin/clavulanate (11.8%) were the most widely prescribed penicillins.

### 2.2. Physicians’ Knowledge of Antibiotics

The correct rate of each antibiotic knowledge question is shown in Table 3. The majority of the surveyed physicians (87.0%) believed that acute URTIs are mainly caused by viral infections (Q1); however, most respondents (77.8%) still chose to prescribe antibiotics for a patient with typical self-limiting URTI symptoms (Q10). Likewise, 66.7% of respondents had difficulty determining correct antibiotic indications (Q9). Nearly sixty percent (59.3%) of respondents confused antibiotics with anti-inflammatory drugs (Q3). Incorrect answers were also likely to appear for the following questions: methicillin-resistant *Staphylococcus aureus* is resistant to beta-lactam antibiotics (Q6), antibiotic restriction policy (Q7), and nephrotoxicity of the first-generation cephalosporins (Q8), with only 28.7%, 31.5%, and 44.4% of respondents giving a correct answer. No significant difference was observed between CHC and THC physicians regarding knowledge of antibiotics.

### 2.3. Determinants of Physicians’ Antibiotic Prescribing Behavior

Multivariate mixed-effects logistic regression analyses with random intercepts for each physician were conducted (Figure 1). The level of physicians’ knowledge regarding antibiotic use for URTIs is significantly associated with their prescribing behaviors. Physicians who scored from 0 to 5 on knowledge assessments were found to be more inclined to prescribe antibiotics (aOR = 1.58, *p* < 0.001), combined antibiotics (aOR = 1.80, *p* = 0.001), and broad-spectrum antibiotics (aOR = 1.56, *p* < 0.001), compared to those who scored from 6 to 10.

Furthermore, a pronounced tendency towards antibiotic prescription was noted within THCs, where antibiotics (aOR = 1.55, *p* = 0.001), combined antibiotics (aOR = 1.89, *p* = 0.001), and broad-spectrum antibiotics (aOR = 1.53, *p* = 0.001) were more likely to be prescribed than in CHCs. Physicians in Xi’an demonstrated a lower likelihood of prescribing combined antibiotics (aOR = 0.47, *p* = 0.002). Female physicians were found to be less inclined to prescribe antibiotics (aOR = 0.70, *p* = 0.015) and broad-spectrum antibiotics (aOR = 0.70, *p* = 0.014) than their male counterparts. Physicians with bachelor’s degrees or higher were less likely to prescribe combined antibiotics (aOR = 0.61, *p* = 0.030).

## 3. Discussion

The empirical findings showed a high antibiotic prescribing rate for acute URTIs in PHC facilities in Shaanxi, China. Broad-spectrum antibiotics were frequently prescribed. A low level of antibiotic knowledge was identified among PHC physicians. Physicians with lower levels of knowledge were more likely to prescribe antibiotics, combined antibiotics, and broad-spectrum antibiotics. Additionally, the type of healthcare facility, geographical region, and physician gender were associated with antibiotic prescribing.

Our study revealed that around sixty percent of sampled URTI visits were prescribed antibiotics. The results were either higher or similar compared to previous studies focused on antibiotic prescription rates for URTIs in China, specifically in Shandong Province (44% in THCs), Hubei Province (52% in CHCs and 67% in THCs), and Guangxi Province (68% in THCs) [16,17,18]. This gap is much more prominent if compared with some developed countries such as Finland (8.8%) and Australia (12.0%) [19,20]. According to the disease-specific quality indicators proposed by the European Surveillance of Antimicrobial Consumption Project (ESAC), the percentage of outpatient URTI visits prescribed antibiotics should not exceed 20% [21]. However, PHC physicians in our study prescribed approximately three-fold more antibiotics than the recommended limit set by the ESAC. The current Chinese guideline also does not recommend routine use of antibiotics for URTIs [22]. Our finding suggests that the overuse of antibiotics is still a serious problem in PHC facilities in Shaanxi Province.

We found that around three-quarters of prescribed antibiotics in our study were broad-spectrum antibiotics. Previous studies indicate that the use of broad-spectrum antibiotics is common in China [6]. However, broad-spectrum antibiotics do not provide better clinical outcomes for self-limiting respiratory tract infections and are associated with an increased risk of AMR [23]. Furthermore, the third-generation cephalosporins, which have a broader antibacterial spectrum than the first- and second-generation cephalosporins, accounted for over one-third of the prescribed antibiotics in our study sample. According to the Chinese guideline, the recommended antibiotic treatment for a URTI with a bacterial cause (i.e., group A streptococcus) is penicillin or first-generation cephalosporins [22]. Our results suggest low adherence to clinical guidelines among PHC physicians in China. A systematic review indicated that many PHC physicians in China did not know how to obtain or use the guidelines, and many guidelines were not applicable to PHC settings [24]. Further training programs are needed to improve PHC physicians’ awareness and adherence to clinical guidelines.

Our analysis revealed that insufficient antibiotic knowledge is significantly associated with antibiotic overprescribing by PHC physicians, which is consistent with a previous study conducted in Spain [12]. Our respondents in CHCs and THCs both demonstrated low levels of knowledge regarding antibiotic use. A significant number of the surveyed physicians believed antibiotics are the conventional treatment for URTIs, and most of our respondents failed to identify correct clinical pointers for antibiotic use. Deficits in knowledge may also lead to uncertainty in antibiotic prescribing [25]. When confronted with uncertainty, prescribing antibiotics can serve as a form of protection or reassurance [26]. A possible reason for insufficient knowledge of antibiotics could be physicians’ low educational background. Nearly half of our respondents did not achieve a bachelor’s degree. Continuous training has been proven to narrow the knowledge gap among physicians with different educational levels and improve their knowledge of antibiotic use [27]. However, current continuous training programs in China may have had little impact on improving physicians’ professional knowledge. A previous study showed that PHC physicians in China were often too busy to attend training, or they found that the training courses were too short and provided insufficient information [28].

Our results showed that other factors may also influence antibiotic use. Firstly, a gender difference was observed, with female physicians prescribing fewer antibiotics than their male counterparts. Similar patterns were also seen in previous studies conducted in China [29]. Secondly, variation in antibiotic prescribing among physicians in different types of facilities and regions was noted. Medical resources are more concentrated in urban regions than in rural or township levels. PHC facilities in less developed areas often lack diagnostic equipment and the expertise of medical staff [30], which might be the reason that the overuse of antibiotics is more common in THCs than in CHCs.

With the increasing concern about antibiotic resistance, effective interventions are required to curb irrational antibiotic use in PHC settings. According to our results, educational interventions targeting physicians’ knowledge regarding antibiotics may have an effective impact on improving antibiotic use. Local administrations should enhance the quality of training for PHC physicians, targeting their low level of antibiotic knowledge and poor awareness of clinical guidelines. Online platforms should be developed to ensure that PHC physicians with heavy workloads can actively participate in training courses. In addition to improving physicians’ professional knowledge, it is also important for physicians to adhere to this knowledge. Behavior-changing interventions such as feedback and social comparison have been proven to optimize physicians’ prescribing behavior [31]. Feedback with individual prescribing data can identify prescribing errors and facilitate physicians to reflect on their prescribing habits, improving their antibiotic prescribing behavior [32]. A previous study showed that a multifaceted intervention including prescription feedback and peer comparison could significantly reduce inappropriate antibiotic prescribing [33]. Furthermore, point-of-care tests should be adequately implemented in PHC settings to reduce physicians’ uncertainty in antibiotic prescribing. Point-of-care tests like C-reactive protein testing have the potential to reduce antibiotic use for acute respiratory tract infections in PHC settings [34]. Lastly, efforts should be made to increase the investment in rural THCs to narrow the resource distribution gap between urban and rural districts, improving healthcare quality for rural residents.

Our study focused on examining the relationship between PHC physicians’ knowledge and their antibiotic prescribing behavior. Unlike some previous studies that relied on self-reported intentions to gauge actual prescribing practices [35,36], we combined prescription data with physicians’ questionnaire surveys. This approach provided a more reliable means to explore the factors influencing antibiotic prescribing. In addition, this is a multi-site survey, and representative areas of the province were selected considering the economic development level (high, medium, and low). The limitations of this study should be acknowledged. Firstly, this study was conducted in only one province. Despite this, the antibiotic prescribing patterns in our study are similar to those in previous studies conducted in China, indicating that the factors contributing to antibiotic overprescribing in our study may also be present in other regions in China with similar socioeconomic levels. Secondly, physicians from village clinics and community healthcare stations were not included due to the lack of an electronic medical record system. A previous nationwide survey showed that only 19% of village clinicians in China had a junior college degree or higher [37]. Therefore, professional medical training is also imperative for village clinicians to improve their clinical knowledge and skills, including rational antibiotic prescribing. Thirdly, since the electronic medical record system in PHC facilities in selected regions did not ensure data integration, we could not include all acute URTI visits during the study period in our analysis. Although a systematic sampling of eligible visits to surveyed physicians was applied, we could not avoid potential selection biases. However, the random sampling process could maximize the representativeness of acute URTI visits included in our study. Lastly, given the cross-sectional nature of the data applied in the current study, we could not establish causality between antibiotic prescribing and the identified factors. Further observational and experimental studies are needed to validate our findings.

## 4. Materials and Methods

### 4.1. Study Design and Sampling

A cross-sectional study was conducted to investigate PHC physicians’ antibiotic prescription behavior for URTI patients and the relationship between the prescription behavior and PHC physicians’ knowledge regarding antibiotics in Shaanxi Province, China. Shaanxi had a population of 39 million and a GDP of CNY 2618.18 billion in 2020, ranking 14th among the 31 provinces in China [38]. PHC physicians included in our study were from community healthcare centers (CHCs) in urban areas and township healthcare centers (THCs) in rural areas. As of 2020, there were 270 CHCs and 1525 THCs in Shaanxi Province, with 5.4 million and 21.1 million patient visits, respectively, throughout the year [39].

We purposely selected three out of ten prefecture-level regions in Shaanxi Province based on their geographic location and economic level as shown in Figure 2. These regions included Xi’an from the central part of Shaanxi, with the highest GDP ranking in the province; Yan’an located in northern Shaanxi, with a GDP ranking of 6th; and Shangluo from southern Shaanxi, with the second-lowest GDP ranking among the ten prefecture-level cities (Appendix A). We then used convenience sampling to recruit PHC facilities in each selected prefecture-level region. Ultimately, 16 CHCs and 14 THCs participated in our study (Appendix A). The physicians who prescribed antibiotics independently in the participating facilities were included in our study.

Within each sampled PHC facility, eligible physicians were recruited to fill out a questionnaire assessing their knowledge of antibiotic use. For physicians who had completed the questionnaire, their prescription data for acute URTI outpatient visits from 1 January 2020 to 31 December 2020 were collected. For each surveyed physician, we randomly selected five URTI visits from the electronic medical record system on the 1st, the 10th, and the 20th day of each month, with a target of 15 visits per month (180 in total during the study period). If the number of visits was insufficient on a sampling day, the sampling would continue on the following day until the number of visits reached 15 in a month. Additionally, if a surveyed physician’s total number of URTI visits did not reach 180 during the study period, all visits were included. Prescription data for sampled visits were linked with questionnaire survey results to analyze the relationship between the knowledge levels of the physicians and their prescribing behavior.

Ethical approval for this study was obtained from the Biomedical Ethics Committee of Xi’an Jiaotong University (reference number 2019-1241). Written consent was sought from all eligible physicians. Patient consent was waived because no contact with patients was conducted and patient anonymity was assured.

### 4.2. Questionnaire Design

The questionnaire included demographic, educational, and occupational basic information of physicians, as well as an assessment of medical knowledge on rational antibiotic prescribing for URTIs consisting of 10 single-choice questions. The questionnaire collected physicians’ personal characteristic information including gender, age, education level, years of working, professional title, location, and the type of healthcare facilities.

Among the ten questions in the knowledge assessment on rational antibiotic use, six questions were based on previous studies in Laos, Vietnam, and China, including the etiology of URTIs, the duration of antibiotic treatment, the spectrum of beta-lactam antibiotics, and questions addressing the need for antibiotics to treat specific URTI symptoms [40,41,42]. We added four more questions: two pertaining to the spectrum and adverse reactions of penicillin and first-generation cephalosporins since they are the recommended antibiotic treatment for bacterial URTIs in the Chinese Guideline [22]; one about the *C. difficile* infection caused by overprescribing of broad-spectrum antibiotics, which were widely prescribed for URTI patients in China [43]; and one question about the antibiotic restriction policy required by the National Health Commission of China [44]. The knowledge questions were reviewed by two infectious disease physicians to assess their relevance and wording. A pilot study was conducted in one PHC facility (not included in the final analysis) to check the physicians’ understanding of the questions.

### 4.3. Definitions of Antibiotics and URTI Visits

Antibiotics were sorted according to the Anatomical Therapeutic Chemical (ATC) classification system [45]. Second-generation to fourth-generation cephalosporins, fluoroquinolones, macrolides, combinations of penicillins, and streptomycins were classified as broad-spectrum antibiotics [46]. Other antibiotics were classified as narrow-spectrum.

Outpatient visits were eligible if they occurred between 1 January 2020 and 31 December 2020, were aged under 80 years, and had a primary diagnosis of a URTI, as defined according to the International Classification of Diseases 10th Revision [47]. We excluded any visits that had a secondary diagnosis of a bacterial infection clearly requiring antibiotics, such as pneumonia, or visits with any severe or chronic disease requiring long-term antibiotic treatment or prophylaxis. Multiple prescriptions for the same patient on the same day in the same facility were considered as one visit in this study.

### 4.4. Data Collection

Each selected facility was visited by two trained investigators from February 2021 to July 2021. The questionnaires were completed without referring to external sources of information. Returned questionnaires were examined by investigators, with missing items being rechecked and refilled immediately on-site.

Prescription data for sampled URTI visits were exported one by one from the electronic medical record system. Prescription details included the generic name of the medicines prescribed, the unique identity number of the prescribing physician (anonymized and used only for identification), and patients’ basic information (age, gender, diagnosis, and payment method).

### 4.5. Data Analysis

Descriptive analyses were used to describe the antibiotic prescribing patterns. Prescribing indicators were the overall proportions of URTI visits that received at least one antibiotic, combined antibiotics, and broad-spectrum antibiotics. The interquartile ranges of these indicators among surveyed physicians were also calculated. The most frequently prescribed antibiotic classes were identified by calculating the proportions of the different agents out of all antibiotic agents that were used.

Physicians’ knowledge about antibiotics was assessed by calculating the total score (ranging from 1 to 10) of the ten antibiotic-related questions in the second part of the questionnaire. The correct rate of each question was compared between physicians in CHCs and THCs using Chi-square tests.

We used mixed-effects logistic regression models with the binary outcomes of whether or not an antibiotic (model 1), combined antibiotics (model 2), and broad-spectrum antibiotics (model 3) were prescribed. In each analysis, a physician was included as a random intercept, while the fixed effects were regions, facility types, physician’s gender, age, education, professional title, working years, and knowledge level of antibiotics. The unit of measure was the patient visit. We also included the patient-level covariates (gender, age, diagnosis, and payment method) in each regression. Adjusted odds ratios (aORs) and confidence intervals were reported. The level of significance was set at *p* < 0.05. All Statistical analyses were conducted in Stata version 16 (Stata Corp, College Station, TX, USA).

## 5. Conclusions

The current study revealed that antibiotic misuse and overuse continue to be a significant problem at PHC facilities in China. Broad-spectrum antibiotics, especially third-generation cephalosporins, are widely used for acute URTIs. Physicians’ knowledge plays a significant role in determining the appropriate use of antibiotics. Professional training targeting physicians’ knowledge of antibiotics and behavior-changing interventions to improve physicians’ adherence to rational antibiotic prescribing is needed. Additionally, the higher antibiotic use in rural settings is concerning. Further studies are needed to understand the underlying reasons for the increased use of antibiotics in rural areas and low socioeconomic settings in China.

## Figures and Tables

**Figure 1 antibiotics-13-00923-f001:**
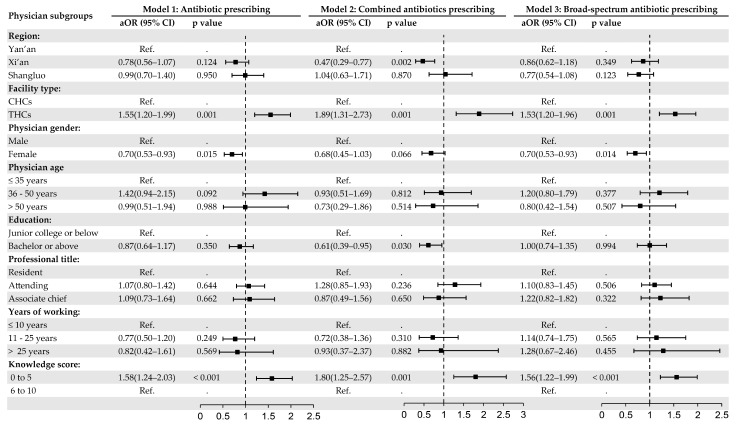
Forest plot of factors associated with antibiotic prescribing patterns for sampled acute URTI visits. Model 1: antibiotic prescribing; model 2: combined antibiotics prescribing; and model 3: broad-spectrum antibiotic prescribing. The adjusted odds ratio (aOR) with 95% CI was obtained from the multivariate binary logistic regressions with random intercepts for each physician. The intraclass correlation coefficient was 0.08 for model 1, 0.12 for model 2, and 0.07 for model 3. CHCs: community healthcare centers. THCs: township healthcare centers.

**Figure 2 antibiotics-13-00923-f002:**
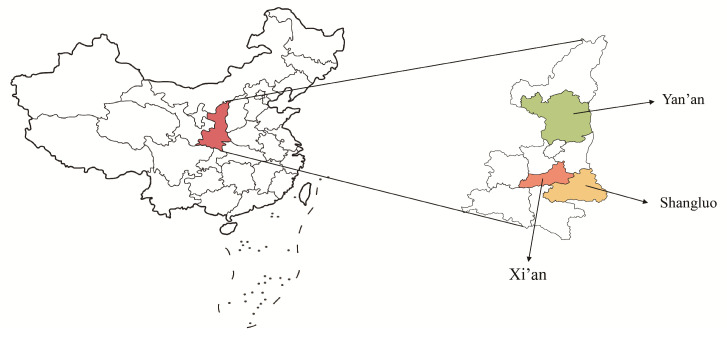
Prefecture-level regions in Shaanxi Province, China, included in this study.

**Table 1 antibiotics-13-00923-t001:** Demographic profiles of 108 surveyed physicians and antibiotic prescribing patterns for 11,217 sampled acute upper respiratory tract infection (URTI) visits across 30 primary healthcare (PHC) facilities in Shaanxi Province, China (2020).

Subgroups	Physicians n (%)	URTI Visits n	Visits Prescribed Antibiotics, % (IQR)	Visits Prescribed Combined Antibiotics, % (IQR)	Visits Prescribed Broad-Spectrum Antibiotics, % (IQR)
Total	108	11,217	61.2 (50.2–72.1)	7.8 (2.3–10.2)	48.3 (36.7–58.7)
**Regions**					
Yan’an	22 (20.4)	2733	61.5 (51.4–68.8)	6.8 (3.5–7.8)	51.3 (42.4–59.0)
Xi’an	41 (38.0)	4235	53.7 (43.3–64.9)	3.7 (1.1–5.8)	44.1 (31.4–58.5)
Shangluo	45 (41.7)	4249	68.6 (59.0–74.7)	12.5 (6.1–11.5)	50.5 (38.5–57.9)
**Facility type**					
CHCs	52 (48.1)	5477	55.5 (41.6–65.7)	5.2 (1.3–7.6)	44.1 (31.0–55.6)
THCs	56 (51.9)	5740	66.6 (56.7–74.9)	10.2 (5.4–11.6)	52.3 (40.8–64.3)
**Physician Gender**					
Male	48 (44.4)	4873	65.9 (58.1–74.2)	10.5 (4.8–11.6)	50.9 (38.7–64.4)
Female	60 (55.6)	6344	57.7 (45.0–68.2)	5.7 (1.8–7.7)	46.3 (34.3–55.5)
**Physician Age**					
≤35 years	31 (28.7)	2974	56.7 (39.8–68.8)	7.6 (2.0–11.4)	43.2 (26.7–52.5)
36–50 years	50 (46.3)	5542	62.7 (51.4–72.2)	6.4 (2.1–8.8)	51.0 (40.4–61.4)
>50 years	27 (25.0)	2701	63.2 (51.2–72.7)	10.7 (2.6–10.6)	48.2 (39.0–58.5)
**Education level**					
Junior college or below	47 (43.5)	4912	64.3 (53.7–72.2)	9.6 (3.7–11.5)	49.3 (39.0–59.0)
Bachelor or above	61 (56.5)	6305	58.8 (48.0–68.8)	6.4 (1.7–8.0)	47.5 (34.1–58.5)
**Professional title**					
Resident	34 (31.5)	3432	63.1 (52.4–72.2)	7.0 (3.3–11.4)	48.5 (40.0–57.9)
Attending	53 (49.1)	5781	59.0 (49.0–72.0)	8.1 (2.1–10.1)	46.8 (31.4–61.4)
Associate chief	21 (19.4)	2004	64.5 (51.2–70.4)	8.0 (3.5–8.7)	51.9 (41.3–58.5)
**Experience**					
≤10 years	25 (23.1)	2311	56.0 (43.3–68.2)	6.4 (2.0–11.4)	40.8 (26.7–49.1)
11–25 years	51 (47.2)	5708	61.8 (51.2–72.3)	6.8 (2.1–8.4)	50.5 (35.8–62.1)
>25 years	32 (29.6)	3198	63.9 (53.1–71.2)	10.5 (3.5–11.2)	49.6 (39.6–58.7)
**Knowledge score**					
0 to 5	59 (54.6)	6407	65.1 (53.7–74.7)	9.4 (3.4–11.4)	51.9 (40.4–62.1)
6 to 10	49 (45.4)	4810	56.1 (49.0–66.2)	5.6 (1.2–8.2)	43.4 (31.4–50.0)

Note: CHCs: community healthcare centers. THCs: township healthcare centers. IQR, interquartile range among physicians.

**Table 2 antibiotics-13-00923-t002:** Categories of antibiotics prescribed for sampled acute URTI visits across 30 PHC facilities in Shaanxi Province, China (2020).

Antibiotic Details	n	Percentage, %
**Total antibiotics prescribed**	7797	
**Spectrum category**		
Narrow-spectrum	1938	24.9
Broad-spectrum	5859	75.1
**Antibiotic subgroups**		
**Third-generation Cephalosporins**	2709	34.7
Cefixime (ATC code J01DD08)	2117	27.1
Cefotaxime (ATC code J01DD01)	251	3.2
**Penicillins**	2303	29.5
Amoxicillin (ATC code J01CA04)	1290	16.5
Amoxicillin/clavulanate (ATC code J01CR02)	919	11.8
**Second-generation Cephalosporins**	858	11.0
Cefuroxime (ATC code J01DC02)	661	8.5
Cefaclor (ATC code J01DC04)	159	2.0
**Macrolides**	663	8.5
Roxithromycin (ATC code J01FA06)	240	3.1
Azithromycin (ATC code J01FA10)	213	2.7
**Fluoroquinolones**	595	7.6
Levofloxacin (ATC code J01MA12)	452	5.8
**Other Antibiotics**	669	8.6

Note: Percentages of specific antibiotics among all antibiotic agents that are used.

**Table 3 antibiotics-13-00923-t003:** Knowledge of 108 surveyed PHC physicians regarding antibiotic use for acute URTIs.

Knowledge Questions	Number (%) of Respondents Giving Correct Answer			
	Total n = 108	CHCs n = 52	THCs n = 56	*p*-Value
Q1: Most acute URTIs are caused by viral infections.	94 (87.0)	43 (82.7)	51 (91.1)	0.38
Q2: It is wrong to stop taking medicine when symptoms begin to improve.	60 (55.6)	25 (48.1)	35 (62.5)	0.32
Q3: Antibiotics are not anti-inflammatory drugs.	44 (40.7)	17 (32.7)	27 (48.2)	0.24
Q4: The overuse of broad-spectrum antibiotics can cause secondary infections like *C. difficile* infection.	103 (95.4)	50 (96.2)	53 (94.6)	1.00
Q5: Penicillin is the antibiotic of choice to treat group A streptococcus infection.	85 (78.7)	41 (78.8)	44 (78.6)	0.67
Q6: Methicillin-resistant *Staphylococcus aureus* is resistant to beta-lactam antibiotics.	31 (28.7)	17 (32.7)	14 (25.0)	0.19
Q7: Only the chief or associate chief physician can prescribe special-use group antibiotics.	34 (31.5)	20 (38.4)	14 (25.0)	0.22
Q8: The first-generation cephalosporins have the highest nephrotoxicity compared with the second-generation and above cephalosporins.	48 (44.4)	19 (36.5)	29 (51.8)	0.24
Q9: Respiratory infectious symptoms, including one of the following symptoms: fast breathing, chest in-drawing, or stridor, should consider antibiotic treatment.	36 (33.3)	17 (32.7)	19 (33.9)	0.30
Q10: For a 6-year-old child who has a fever of 38 °C, purulent nasal discharge, and sore throat for two days, symptomatic treatment is recommended.	24 (22.2)	14 (26.9)	10 (17.9)	0.64
**Total knowledge score (Mean ± SD)**	5.2 ± 1.8	5.2 ± 1.8	5.3 ± 1.7	0.45

Note: The *p*-values derived from Chi-square (Fisher’s exact) tests or Wilcoxon–Mann–Whitney tests. CHCs: community healthcare centers. THCs: township healthcare centers.

## Data Availability

All data and material are available from the corresponding author upon reasonable request.

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
