# Peer review of "Primary Healthcare Physicians’ Insufficient Knowledge Is Associated with Antibiotic Overprescribing for Acute Upper Respiratory Tract Infections in China: A Cross-Sectional Study"

_antibiotics, 2024, doi:10.3390/antibiotics13100923_

Round 1
Reviewer 1 Report
Comments and Suggestions for Authors
It was a delight to read the manuscript “Primary Healthcare Physician’s Insufficient Knowledge is Associated with Antibiotic Overprescribing for Acute Upper Respiratory Tract Infections in China: A cross-sectional study”. The study is exceptionally well performed and of high interest. There are a few aspects that in my humble opinion would make a good manuscript even better:
A) The mean survey score is stated to be 5.24. This precision is too high given that there were only 108 physicians. Please see: https://science-network.tv/significant-figures/. Hence, I recommend writing 5.2 and not 5.24. Subsequently, I recommend the response rate to be written as 91% and not 90.8% (and all other percentages related to physicians should be presented with only two significant figures).
B) In table 1 there is a good presentation of various descriptive statistics. However, for each area (and for total of all areas) I would like to also see the interquartile range or min and max values for “Visits prescribed antibiotics”. This gives a brief estimate of variability and that is interesting to see. Other studies suggest the variability is a factor of 5-10 (perhaps minimum proportion of visits prescribed antibiotics in one area would for one physician be 7% and the maximum for another physician be 70%).
C) The main conclusion is that “Professional training targeting physicians’ knowledge of antibiotics are urgently needed”. Previous studies implies that the outcome of such training is varying. The authors themselves concludes “The majority of the surveyed physicians (87.0 %) believed that acute URTIs are mainly caused by viral infections (Q1); however, most respondents (77.8 %) still chose to prescribe antibiotics for a patient with typical self-limiting URTI symptoms”. This finding implies that lack of information may not be the main problem but rather adherence to the knowledge physicians have. Hence, other actions might also be relevant such as letting pharmacists manage some minor illnesses such as uncomplicated URTI. I am not sure if trained pharmacists are widely available in China but this has been trialled in other parts of the world with good results (a reduction in antibiotic prescribing). Could the authors mention this as a possible action alongside with further education of physicians?
Reviewer 2 Report
Comments and Suggestions for Authors
General comments:
This is a very interesting and necessary study. However, the sample size is insufficient and perhaps the conclusions are somewhat pretentious.
.- Material and methods. Some doubts arise. An estimation of the sample size was made? Why were the 3 groups of physicians not more homogeneous, e.g. 30-30-40?
Good methodological design. Results correctly presented and discussion resolved effectively including limitations section, congratulations. A specific subsection on strengths of the study is missing.
.- References. 43 quotes are included of which 21 are recent (49%), 5 years or less from their publication
Specific comments:
Table 2. Perhaps it could be presented as a bar diagram. However, in the current format it is correct. Consider including any additional recent references.
